# Transscleral Fixation of Black Diaphragm Intraocular Lens in Complete Aniridia and Aphakia Due to Posttraumatic Eye Rupture: A Pilot Study

**DOI:** 10.3390/jcm8010046

**Published:** 2019-01-05

**Authors:** Tomasz Chorągiewicz, Katarzyna Nowomiejska, Dariusz Haszcz, Dominika Nowakowska, Teresio Avitabile, Michele Reibaldi, Anselm Gerhard Maria Jünemann, Mario Damiano Toro, Robert Rejdak

**Affiliations:** 1Department of General Ophthalmology, Medical University of Lublin, ul. Chmielna 1, 20-079 Lublin, Poland; tomekchor@wp.pl (T.C.); katarzynanowomiejska@umlub.pl (K.N.); haszcz@umlub.pl (D.H.); dominika.nowakowska85@gmail.com (D.N.); robertrejdak@yahoo.com (R.R.); 2Institute for Ophthalmic Research, University Eye Hospital, 72076 Tübingen, Germany; 3Eye Clinic, University of Catania, Via S. Sofia 78, 95123 Catania, Italy; t.avitabile@unict.it (T.A.); mreibaldi@libero.it (M.R.); 4Department for Ophthalmology, University Medicine Rostock, Doberaner Straße 140, 18055 Rostock, Germany; Anselm.Juenemann@med.uni-rostock.de; 5Department of Experimental Pharmacology, Medical Research Centre, Polish Academy of Sciences, 02-106 Warsaw, Poland

**Keywords:** eye rupture, vitrectomy, aniridia, aphakia, intraocular lens

## Abstract

Introduction: To assess long-term outcomes of implantation of black diaphragm intraocular lens (BD IOL) in post-traumatic aniridia and aphakia due to eye rupture. Methods: This is a retrospective consecutive case series of 14 eyes with post-traumatic complete aniridia and aphakia treated with scleral fixation BD IOL. Measurements included ophthalmological comorbidities, best corrected visual acuity (BCVA), complications, and postoperative interventions. The average postoperative follow-up period was 36 months. Results: BCVA improved in 6 cases, was stable in 6 cases and worsened in 2 cases. The lens was well centered in 13 cases. Glaucoma was diagnosed in six cases developed, and three of them required Ahmed valve implantation. One lens developed opacity. The cornea was decompensated in 6 cases, while two of them required penetrating keratoplasty. Conclusion: Implantation of BD IOL in eyes with severely traumatized eyes enables reconstruction of the anterior segment and some functional restoration, although many complications may arise during the longitudinal follow-up.

## 1. Introduction

Eyeball rupture is the most devastating type of open-globe injury caused by a blunt object leading to sudden increase of intraocular pressure and disruption of the wall of the eye globe [1]. It often results in both aniridia and aphakia, in addition to vitreous haemorrhage and retinal detachment [2]. This condition is usually primarily treated by the corneal and scleral suturing, followed by secondary lens extraction, pars plana vitrectomy (PPV), and retinal reattachment to keep the eyeball complete.

In addition to anatomical failure in eyes with posttraumatic aniridia and aphakia, different functional phenomena occur, as photophobia, glare, epiphora, and blurred vision. Moreover, cosmetic defect might bother the patient, as this usually mostly occurs in patients of a young age.

Different therapeutic approaches have been proposed to eliminate these problems, such as corneal stromal implants, pupilloplasty, colored contact lenses, and even corneal tattooing [3,4]. Another problem is lack of support from the iris, so it is difficult to position intraocular lens (IOL) in the bag or sulcus. Thus, there are difficulties with performing conventional surgery with conventional IOLs in the case of aniridia and aphakia.

To overcome these problems, a posterior chamber IOL with an opaque peripheral segment to simulate the iris diaphragm has been developed by Morcher and reported for the first time in 1994 by Sundmacher in congenital aniridia [5,6]. It can be implemented to the sulcus or scleral fixated. In early designs, the haptics were found to be too fragile and the IOL too large for easy implantation. These problems have been addressed by modifications to subsequent models [7].

In the current study, we describe our experience in a series of 14 cases with posttraumatic aniridia and aphakia due to eye rupture managed with secondary implantation of the BD IOL. Primary wound closure with lens extraction has been performed before or the lens was lost during trauma.

## 2. Material and Methods

This is a retrospective study of 14 consecutive patients who underwent black diaphragm intraocular lens (BD IOL) implantation at the Department of General Ophthalmology of Medical University of Lublin in a time period from 1 January 2011 to 1 January 2017. This study followed the tenets of the Declaration of Helsinki. The treatment chosen in the study was a part of a standard care. Written informed consent was obtained from all subjects. Local Ethics Committee approval has been given.

Inclusion criteria were as follows: (1) complete aniridia and aphakia due to eye rupture, (2) excessive photophobia, (3) and at least 12 months of the follow-up. Exclusion criteria included active ocular inflammation or infection. Medical records of patients were reviewed retrospectively (Table 1).

The data collected included: pre- and postoperative best corrected visual acuity (BCVA), slit lamp evaluation, applanation tonometry, and fundus examination. BCVA was determined using Snellen charts; hand movement (HM), finger counting (FC), light perception (LP), and no light perception (NLP) forms were used to describe visual acuities of included patients, if they were less than 0.1. Intra- and postoperative complications were collected to evaluate the safety of the procedure.

Most patients were men (12 patients). Mean patient age was 43 years (range 25 to 79 years) and the mean follow-up—36 months (range 12 to 120 months). Patients were examined 1 week, 1 month and 3 months postoperatively, and then at 3-months intervals, more often if indicated. Postoperative medication included topical dexamethasone and levofloxacin 5 times daily for one month.

### 2.1. Black Diaphragm Intraocular Lens

BD IOL is a poly-methyl methacrylate (PMMA) IOL (Morcher, GmBH, Stuttgart, Germany) with an overall diameter of 12.5 mm (67 G and 68) or 13.5 mm (67 F and 67 L). Different types of BD implants were used: 67 F, 67 G, 67 L, and 68. Type 67 L has an opening at 6 o’clock in the diaphragm and it is dedicated for vitrectomized eyes. Type 67 F is for myopic eyes, and 67 G is available without optics. Type 68 has a smaller optic diameter (4.5 mm instead of 5 mm). Diopter range is available from +10 to +30 diopters for all types. The power of the lens was calculated according to the SRK (Sanders, Retzlaff and Kraff) II formula.

Each end of the C-shaped haptics has an eyelet for suture fixation. The 10.0 mm optic presented a 3.0 mm (67 B) or 5.0 mm (67 G) clear central zone surrounded by a peripheral diaphragm of black PMMA.

### 2.2. Surgical Technique

The surgeries were done in general (5 cases) or peri-bulbar (9 cases) anaesthesia. BD IOL was implemented by one surgeon (DH) in all cases.

The procedure of BD IOL implantation was similar to those previously described [5,6]. In 7 cases, anterior chamber maintainer was used and for the remaining 7 cases PPV with constant infusion was performed prior to BD IOL implantation.

All the patients underwent transscleral fixation of BD IOL. First, a scleral tunnel was created 10 mm long posteriorly to the superior limbus (Figure 1) by creating a scleral triangle flap (Figure 2). At 6 o’clock another scleral flap was made (Figure 3). Then a 3.2 mm keratome knife was used in the superior scleral tunnel. Thereafter, a double-armed 10-0 Prolene suture was passed vertically (perpendicular to sclera) 2 mm behind the surgical limbus with the straight needle into the anterior chamber under the 6 o’clock scleral flap and engaged in the bore of a 27-gauged bent needle introduced and pulled out under the 12 o’clock scleral flap (Figure 4). Similarly, another 10-0 Prolene was introduced 2 mm vertically apart from the first suture. The two sutures were brought out from the surgical wound and tied to the eyelets of the BD IOL (Figure 5) and the BD IOL was implanted while pulling the two sutures from the side (Figure 6). Finally, the superior scleral tunnel was sutured after IOL insertion (Figure 7) and conjunctiva was sutured using Vicryl 8.0 suture.

In two cases of simultaneous penetrating keratoplasty, BD IOL was implanted through the corneal trephination.

### 2.3. Patient and Public Involvement

Patients and or Public Were Not Involved.

## 3. Results

Visual acuity improved in 6 cases, remained the same in 6 cases and worsened in the remaining 2 cases. Preoperatively, BCVA was no light perception in one eye, light perception in 4 eyes, hand movement in 4 eyes, counting fingers in 1 eye and 0.02 in one eye, 0.1 in one eye, 0.2 in two eyes. Postoperatively, it was light perception in four eyes, hand movement in two eyes, counting fingers in four eyes, 0.02 in one eye, 0.1 in one eye, 0.2 in one eye, and 0.5 in one eye (Table 2).

Before surgery, 13 eyes were aphakic and 1 eye was pseudoaphakic.

Seven patients had anterior vitrectomy and seven had PPV due to coexisting retinal detachment. In six cases, 5000 Cst silicone oil was used as a tamponade, while in one case C3F8 gas was used. Simultaneous penetrating keratoplasty (PKP) was performed in 2 cases. In one case previously implanted IOL was removed, due to opacity on its surface.

All surgeries were completed uneventfully. Mild anterior chamber inflammatory reaction was resolved within 1 month after BD IOL implantation. Preoperatively, there was increased IOP in 2 cases (14%). Ahmed valve implantation was necessary in three cases (cases 5, 8, and 13). Glaucoma was controlled with medication in another 3 cases (cases 5, 7, and 12). Overall, glaucoma was present in 6 eyes postoperatively (43%). Mean age of patients with or without glaucoma was 36.5 and 48 years respectively. Hypotony (IOP less than 10 mmHg) was observed in 4 cases (cases 1, 2, 6, and 9)—28% (Table 2).

The cornea was decompensated in 6 cases postoperatively (42%). PKP was performed simultaneously in two eyes (cases 7 and 8), but graft opacities evolved in those patients. PKP was needed postoperatively in 2 cases (cases 5 and 9).

BD IOL subluxation was observed in one case (case 12) and it was repositioned as a second procedure.

Postoperatively, silicone oil in the anterior chamber was found in 3 cases (cases 1, 2, and 4). Two cases needed further vitrectomy with silicone oil exchange (cases 5 and 6) due to retinal redetachment. All eyeballs were preserved. All the patients observed diminishing of glare and photophobia, as well as good cosmetic effect.

## 4. Discussion

In the present study we reported the long-term clinical outcomes of implantation of BD IOL in patients with both aniridia and aphakia as a results of post-traumatic eye rupture. It is expected that BD IOL can reduce the glare, enhance the aesthetic function, and improve the visual acuity in these patients.

Good clinical effects of BD IOL implantation have been described so far in both posttraumatic and congenital aniridia [8,9]. However, the risk of complications is quite high after BD IOL implantation [9]. Short term complications include difficulties in implantation, choroidal detachment, and persistent ocular inflammation. In our series, we observed persistent ocular inflammation, but it was controlled well with topical steroid drops.

Glaucoma is the most common complication of BD IOL implantation. In our study, it was found in 43% postoperatively (14% preoperatively) and did not correlated with the age of patient, while in other studies it was observed from 40 to 75% [8,9,10,11,12,13]. It is supposed that the reason for increased intraocular pressure is direct compression from the haptics to the trabecular meshwork [7,14]. Moreover, substantial damage to the anterior segment may also play a role in glaucoma after BD IOL implantation.

Corneal opacities were seen in 6 cases of our patients and were due to the decline in the number of corneal endothelial cells. Bullous keratopathy, resulting in visual loss and long-term pain, was seen in two patients, and thus for further management PKP was required. The most possible reason for these complications is the fact that BD IOL is larger and heavier than a standard scleral-fixated IOL. Moreover, movement of the BD IOL with postural changes and contact of BD IOL edge with the anterior chamber angle and cornea causes corneal endothelial loss and damage of the blood-aqueous barrier. Some modifications have been developed to avoid this situation, such as adding two sutures in front of the BD IOL to limit the space for rotation [15]. Corneal epithelial cells might be lost during surgical procedure due to effect of PPV or anterior chamber infusion in aphakic and aniridic eyes. To prevent endothelial cells loss during operation, intraoperative IOP control and protection with dispersive ophthalmic viscoelastic devices should be applied.

The largest study so far was a study performed by Qiuand et al. [16]. The authors reported outcomes of demographics and clinical evaluation in 95 patients. 38% of patients developed glaucoma postoperatively, and corneal decompensation occurred in 9% of eyes.

BD IOL has been used so far not only in posttraumatic, but also in congenital aniridia [16]. Qiu and colleagues described 23 eyes of congenital aniridia. Glaucoma and corneal decompensation were major complications of this procedure. However, several complexities are related to congenital aniridia, such as nystagmus, limbal stem cell deficiency, and a higher risk of developing glaucoma.

The visual acuities in our series of 14 patients were quite poor, especially in eyes with accompanying retinal detachment and simultaneous PPV. However, effective visual rehabilitation was obtained by reducing symptoms such as glare and photophobia in patients with aniridia and aphakia.

The first reason for poor postoperative visual acuity is the condition of the retina, as BD IOL implantation was combined with PPV due to coexisting posttraumatic retinal detachment in half of the cases in our series. The second factor is substantial damage to the anterior segment due to extensive trauma. The third point is astigmatism induced by a large (10 mm) scleral incision, as the diameter of BD IOL is large, and additionally by potential tilting of the implant without sufficient support and fixation.

The follow-up period in our study (36 months) is quite long. It is comparable to 41 months in the study done by Dong [9], 25 months in the study performed by Qiu [16], and 33 months in the study done by Beltrame [8].

The shortcomings of the present study are: a lack of postoperative ultrasound biometry measurement examination confirming the localization of the BD IOL and a lack of endothelial cell density.

In conclusion, BD IOL can be used for simultaneous anatomical reconstruction of the anterior segment and functional improvement in eyes after severe eye trauma. BDI IOL implantation is a sophisticated surgery and there is a need for highly skilled surgeons, as there is a high risk of complications. A long-term and steady follow-up period of clinical course is required to monitor patients with BD IOL postoperatively. Moreover, inclusion criteria for patients should be very narrow with a meticulous preoperative check-up.

## Figures and Tables

**Figure 1 jcm-08-00046-f001:**
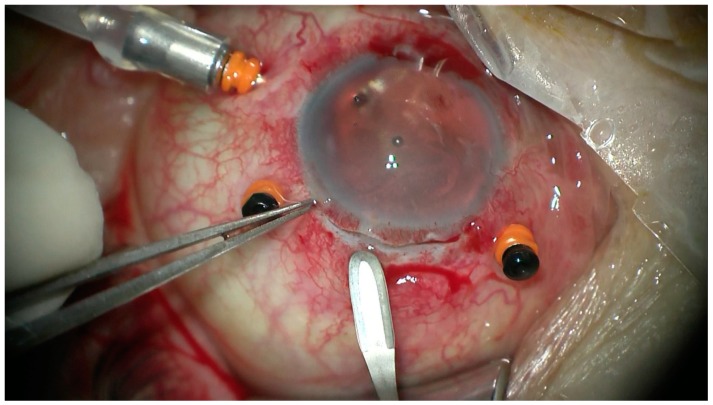
Creating a scleral tunnel 10 mm long after cutting the conjunctiva.

**Figure 2 jcm-08-00046-f002:**
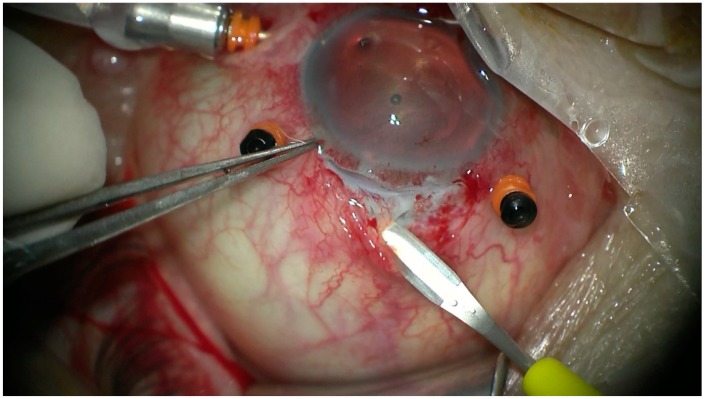
Performing a scleral triangle flap.

**Figure 3 jcm-08-00046-f003:**
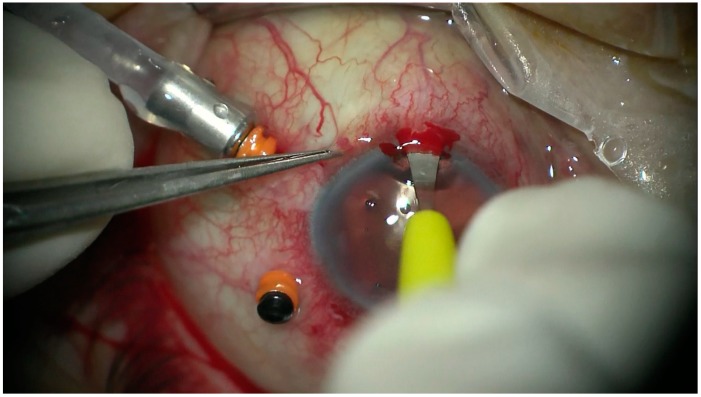
Performing the second scleral flap at 6 o’clock.

**Figure 4 jcm-08-00046-f004:**
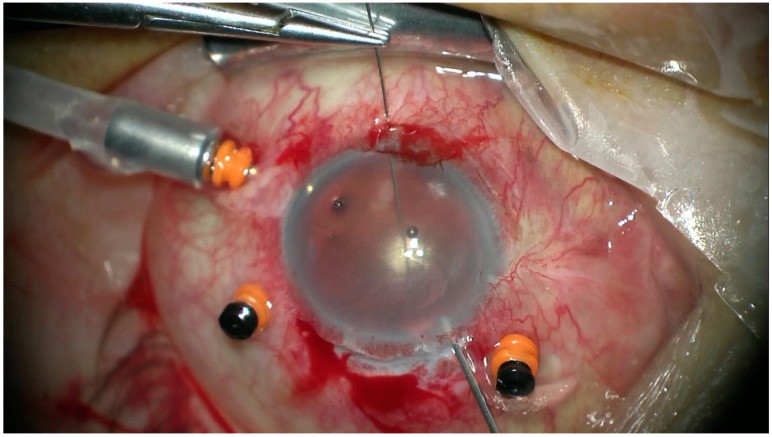
Passing 10-0 proline suture vertically with the straight needle into the anterior chamber under the 6 o’clock scleral flap and engaging in the bore of a 27 gauged bent needle introduced pulled out under the 12 o’clock scleral flap.

**Figure 5 jcm-08-00046-f005:**
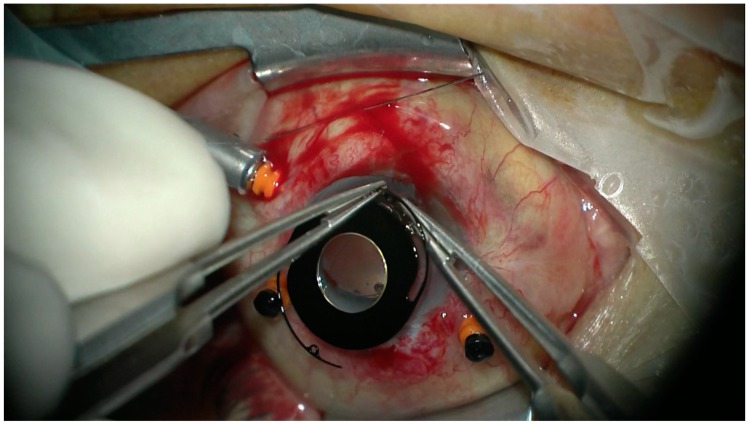
Tying the sutures to the eyelets of the black diaphragm intraocular lens.

**Figure 6 jcm-08-00046-f006:**
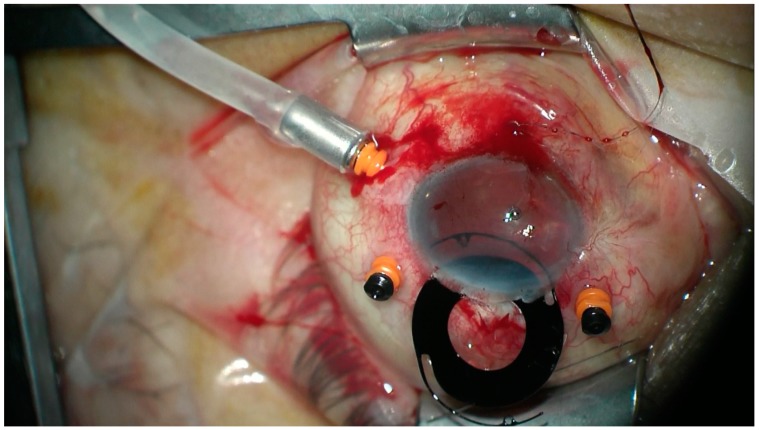
Implanting of black diaphragm intraocular lens while pulling the two sutures from the side.

**Figure 7 jcm-08-00046-f007:**
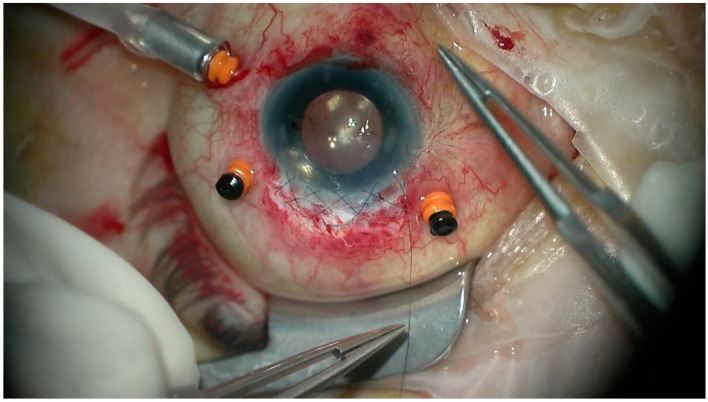
Suturing the superior scleral tunnel after black diaphragm intraocular lens insertion.

**Table 1 jcm-08-00046-t001:** Status of 14 patients monitored after black-diaphragm intraocular lens (BD IOL) implantation: gender, age, initial and final best-corrected visual acuity (BCVA), postoperative intraocular pressure, ocular comorbidities, surgical procedure, type of BD IOL, postoperative complications and secondary interventions. Abbreviations: PPV-pars plana vitrectomy, PKP-penetrating keratoplasty.

Number	Gender	Age	Follow Up Period (Months)	Initial BCVA	Final BCVA	Final IOP (mmHg)	Ocular Comorbidities	Surgical Procedure	Type of BD IOL	Postoperative Complications	Secondary Interventions
1	male	57	14	light perception	counting fingers	4	retinal detachment, vitreous haemorrhage	BD IOL scleral fixation + PPV + silicone oil	68	silicone oil in the anterior chamber, hypotony	None
2	male	49	13	no light perception	light perception	5	retinal detachment	BD IOL scleral fixation + PPV + silicone oil	67 L	silicone oil in the anterior chamber, hypotony	None
3	male	31	13	light perception	light perception	22	corneal scar	BD IOL scleral fixation + anterior vitrectomy	67 G	corneal eodema, glaucoma	None
4	male	30	24	temporal light perception	counting fingers	18	retinal detachment, vitreous haemorrhage, corneal scar	BD IOL scleral fixation + PPV + silicone oil	76 L	corneal scar, silicon oil in anterior chamber, BD IOL opacity	None
5	male	43	24	light perception	counting fingers	15	retinal detachment, vitreous haemorrhage, corneal scar, glaucoma	BD IOL scleral fixation + PPV + silicone oil	67 L	corneal edema, glaucoma	PKP + PPV + silicone oil exchange; Ahmed valve implantation
6	male	25	32	hand movement	light perception without localization	6	retinal detachment, vitreous hemorrhage, corneal scar	BD IOL scleral fixation + PPV + silicone oil	67 G	Hypotony	PPV + silicone oil exchange × 2
7	female	35	61	counting fingers	light perception	12	corneal scar	PKP + PC IOL removal + BD IOL scleral fixation + PKP	67 F	glaucoma, corneal opacity	None
8	male	43	49	hand movement	hand movement	18	corneal opacity, glaucoma	PKP + BD IOL scleral fixation + anterior vitrectomy + PKP	67 G	graft disease, glaucoma	Ahmed valve implantation
9	female	49	21	hand movement	counting fingers	5	vitreous hemorrhage, corneal scar, post traumatic cataract, macular scar	BD IOL scleral fixation + anterior vitrectomy +	67 G	corneal graft opacity, hypotony	PKP
10	male	43	30	0.1	0.1	12	corneal scar	BD IOL scleral fixation + anterior vitrectomy	76 G	non	None
11	male	50	12	hand movement	hand movement	14	retinal detachment, strabismus	BD IOL scleral fixation + PPV + silicone oil	67 G	none	None
12	male	41	55	0.2	0.5	19	corneal scar, post traumatic cataract	BD IOL scleral fixation + anterior vitrectomy	67 G	BDI IOL subluxation, glaucoma	BD IOL repositioning
13	male	26	42	0.02	0.02	18	retinal detachment, corneal scar	BD IOL scleral fixation + PPV + C3F8	67 G	glaucoma, corneal opacity	Ahmed valve implantation
14	male	79	120	0.2	0.2	18	corneal scar	BD IOL scleral fixation + anterior vitrectomy	67 G	none	None

**Table 2 jcm-08-00046-t002:** Demographics, best corrected visual acuity (BCVA) and IOP results.

Gender		Age			BCVA				BCVA Change (Patients)			Final IOP (mmHg)		
Male	Female	Min.	Mean	Max.	Initial		Final		Deterioration	No Change	Improvement	Min.	Mean	Max.
12	2	25	43	79	**Min.**	**Max.**	**Min.**	**Max.**	2	6	6	4	13.3	22
					no light perception	0.2	light perception	0.5						

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
