# Peer review of "Transscleral Fixation of Black Diaphragm Intraocular Lens in Complete Aniridia and Aphakia Due to Posttraumatic Eye Rupture: A Pilot Study"

_jcm, 2019, doi:10.3390/jcm8010046_

Reviewer 1 Report

The authors in the manuscript have summarized the findings of a retrospective study to assess long-term outcomes of implantation of black diaphragm intraocular lens in posttraumatic aniridia and aphakia due to eye rupture. In their longitudinal follow-up of the patients, reconstruction of the anterior segment and functional restoration of traumatic eyes was also associated with increase risk of other complications such as glaucoma, corneal opacities etc.

Comments:

1.     Lines 39-40: provide appropriate references for therapeutic approaches.

2.     Line 51, Text is incomplete.

3.     Table 1, gender, age, type of trauma are not mentioned in the table, please include these parameters in table.

4.     Lines 136-141: As glaucoma incidence is higher with aging, it is important to mention the association of glaucoma with age.

5.     Results can also be summarized in table format.

Author Response

Dear Reviewer,
         thank You for Your precious comments that let us to improve greatly the quality of our paper. We reviewed the manuscript as You recommended.

Sincerely,

Prof. Dr. Michele Reibaldi

Editor Section of Ophthalmology Journal of Clinical Medicine

Reviewer 2 Report

I have the following points for the authors to address:

1.      Line 15, delete extra space before “complete.

2.      Line 16, change “treated with scleral fixation of BD IOL with” to “treated with scleral fixation BD IOL.”

3.      Line 23, change “functional restoration” to “some functional restoration”.

4.      Line 35 to line 37, delete the sentence “They are caused by……iris defects.”, an aphakic eye has no crystalline lens.

5.      Line 42 to line 43, “withperforming” should be changed to “with performing” and “cataract” should be deleted in the sentence because it is aphakia.

6.      Line 91 and 92, how many patients were used by AC maintainer and how many patients were performed by PPV infusion?

7.      Line 95, “Inferior” limbus should be “superior” limbus. This change will make the statement consist with superior tunnel described later (line 97 and figure 7 legend). (Since the patient is oriented opposite to the surgeon, superior is at 12 o’clock of patient and 6 o’clock of surgeon.)

8.      Line 96, “flapwas” should be “flap was”.

9.      Line 99, “andengaged” should be “and engaged”.

10.   Figure 4 legend, “andengaging” should be “and engaging”.

11.   Figure 5 legend, “Tieingthe” should be “Tieing the”.

12.   Line 171, corneal endothelial loss might due to operation procedure (such as PPV infusion in aphakic and aniridia eye) as well. It would be better if the authors had data on corneal endothelial cell count measurements before and after the operation procedure. The authors had better to discuss this and give suggestion of endothelial protection method.

13.   Line 184, astigmatism usually will not cause further large amount of visual acuity loss in an eye with poor vision. It would be better to put the discussion of astigmatism at the end of the paragraph. Other than caused by large incision, astigmatism can also be caused by tilted BD IOL due to iridia, to discuss better lens design and suture location would be a plus.

14.   Line 189, “to41” should be “to 41”.

15.   Line 192, does “UBM” stand for “ultrasound biometry measurement”?

Author Response

(The authors gave the same response as above.)
